# Racial Identity-Aware Facial Expression Recognition Using Deep Convolutional Neural Networks

Muhammad Sohail [1]![ID], Ghulam Ali [1,*]![ID], Javed Rashid [2,3,*]![ID], Israr Ahmad [4]![ID], Sultan H. Almotiri [5], Mohammed A. AlGhamdi [5]![ID], Arfan A. Nagra [6] and Khalid Masood [6]

1. Department of CS, University of Okara, Okara 56310, Pakistan; sohailm816@gmail.com
2. Department of CS&SE, Islamic International University, Islamabad 44000, Pakistan
3. Information Technology Services, University of Okara, Okara 56310, Pakistan
4. Department of Computer Science and Technology, Chongqing University of Posts and Telecommunications, Chongqing 400065, China; israrcsc5@gmail.com
5. Computer Science Department, Umm Al-Qura University, Makkah City 21961, Saudi Arabia; shmotiri@uqu.edu.sa (S.H.A.); maeghamdi@uqu.edu.sa (M.A.A.)
6. Department of Computer Science, Garrison University, Lahore 94777, Pakistan; arfan137nagra@gmail.com (A.A.N.); kmasoodk@gmail.com (K.M.)
* Correspondence: GhulamAli@uo.edu.pk (G.A.); RanaJavedRashid@gmail.com (J.R.); Tel.: +92-301-6010309 (G.A.); +92-331-2627786 (J.R.)

**Abstract:** Multi-culture facial expression recognition remains challenging due to cross cultural variations in facial expressions representation, caused by facial structure variations and culture specific facial characteristics. In this research, a joint deep learning approach called racial identity aware deep convolution neural network is developed to recognize the multicultural facial expressions. In the proposed model, a pre-trained racial identity network learns the racial features. Then, the racial identity aware network and racial identity network jointly learn the racial identity aware facial expressions. By enforcing the marginal independence of facial expression and racial identity, the proposed joint learning approach is expected to be purer for the expression and be robust to facial structure and culture specific facial characteristics variations. For the reliability of the proposed joint learning technique, extensive experiments were performed with racial identity features and without racial identity features. Moreover, culture wise facial expression recognition was performed to analyze the effect of inter-culture variations in facial expression representation. A large scale multi-culture dataset is developed by combining the four facial expression datasets including JAFFE, TFEID, CK+ and RaFD. It contains facial expression images of Japanese, Taiwanese, American, Caucasian and Moroccan cultures. We achieved 96% accuracy with racial identity features and 93% accuracy without racial identity features.

**Keywords:** deep learning; cross culture; facial expression; convolution network; racial identity

## 1. Introduction

The aim of a facial expression recognition system is to recognize the discrete categories of facial expressions such as happy, sad, surprise, angry, neutral, fear, contempt and disgust from still images or video sequences. Though several recent studies focus on image sequence based facial expression recognition tasks [1], still image based facial expression recognition remains a difficult problem due to the following three reasons. First, the facial structure variations among the subjects of different cultures thus make the classification task difficult in some cases [2]. Second, inter-expression resemblance between some expressions might be significant and thus challenging to recognize them accurately [3]. Third, different subjects express their emotions in different ways due to facial appearance variation and facial biometric shapes [4]. These three challenging issues can be visualized in Figure 1. It contains five representative faces, labeled as "Moroccan", "Caucasian", "Taiwanese", "American", "Japanese" cultural, ethnic and geographic region, respectively.

However, different subjects have great variations in facial expression representations due to inter-expression resemblance, facial structure and facial appearance.

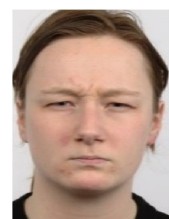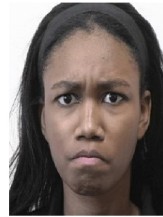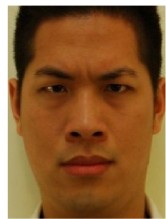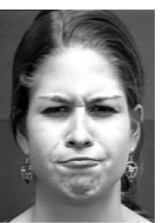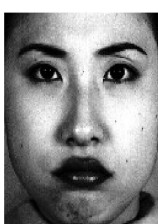

**Figure 1.** Facial expression representation variations from Moroccans, Caucasians to Taiwanese, Americans, and Japanese subjects.

Generally, the cross-culture variability of facial structure could possibly lead to incorrect recognition of facial expression because the facial image of an emotion from one culture could be very dissimilar to facial images of the same emotion from other cultures (e.g., the facial images of "angry" emotion in Figure 1). Furthermore, psychological studies show that facial expression representation varies from culture to culture, and subjects from different cultures express different arousal levels to present facial expressions [5]. Therefore, we believe that, with the inclusion of a compact description of racial identity as an auxiliary input to the model, the facial expression recognition system becomes more reliable and robust against cross-culture variations just as the role of culture identity in speech recognition systems [6].

Convolutional neural networks outperformed the traditional approaches and even human beings in other fields such as plant diseases identification [7], IoT based smart parking system [8], Brain Tumor Segmentation [9], and so on facial expression recognition [10]. Moreover, in cross-culture facial expression recognition process, there is still a need to enhance the performance of a cross-culture facial expression recognition system. Intra-culture and inter-culture expressions' variability, unreliable expression labels, and lack of a large-scale multi-culture facial expression dataset restrain the performance of cross-culture facial expression recognition techniques. Therefore, in this study, we developed a multi-culture facial expression dataset to jointly learn the cultural identity and facial expression for cross-culture facial expression recognition.

In this paper, a novel joint deep learning technique is developed to recognize the racial identity aware facial expressions. The proposed racial identity-aware network (RIA-Net) learns the facial expressions from facial images and extracts the racial identity features from a pre-trained racial identity network (RI-Net). The RI-Net is trained using the multi-culture dataset with culture annotations such as Japanese, Taiwanese, American, Caucasian and Moroccan. The fusion of expression features and racial identity features makes the model robust to the racial identity variations. Therefore, this study makes a major contribution to research on multi-culture facial expression recognition by learning the cultural variations in facial expression representation. To the best of our knowledge, there is no literature work using a racial identity feature with expression features merged to jointly learn the multi-culture facial expression representation.

In the following sections, we first provide an overview of related works in Section 2. The proposed model architecture and overview of the dataset used in this paper are explained in Section 3. We then provide the experimental results and model performance visualisation in Section 4. Finally, we conclude the paper in Section 5.

## 2. Related Work

It has been proved that facial expression representation is not only influenced by muscular deformation of facial structure, but also by many other social factors such as culture, geography, and ethnicity [11]. Most of the studies in the area of facial expression recognition research have investigated only the effects of muscular deformation of facial

structure [12]. However, the social factors are ignored, with very few studies addressing it so far. Ali et al. [13] present a multi-cultural face expression detection method based on enhanced NNE (neural network ensemble) collections. The acted still images were gathered from three databases: JAFFE, TFEID and RadBoud, and came from four distinct cultural and ethnic groups. Srinivasan and Martinez [14] presented the visual analysis of cross-culture and culture-specific representation and perception of facial expressions.

Convolution neural network (CNN) based models have been widely used for racial identity recognition and facial emotions recognition. Vo et al. [15] compared the performance of two deep learning models, the RR-CNN and RR-VGG, for race recognition of Japanese, Chinese, Brazilian and Vietnamese people. Similarly, Wang et al. [16] tackle the problem of ethnicity recognition using deep CNN. The proposed CNN model classifies the sample subjects as white or black, Chinese or Non-Chinese, Han and Uyghur or Non-Chinese. Moreover, deep learning is currently an emerging area of research that significantly improves the performance of facial expression recognition systems. One of the current trends in facial expression recognition is the use of multiple deep convolutional neural networks. Pons and Masip [17] proposed two stages of deep learning architecture to recognise facial expressions. First, a set of deep convolutional neural networks trained to learn facial expressions from sample images and applied DCNN at the second stage to combine individual DCNN decisions to predict the presence of an expression. Wen et al. [18] used a set of CNN's as base classifiers to output the probability of each facial expression and then fused these probabilities through the probability-based fusion technique to classify the facial expressions. Shim et al. [19] trained a multitasking deep convolutional neural network model for facial region detection and face angle estimation, along facial expression classification.

The most relevant work to this study is presented in [20], where joint learning is proposed to recognise facial expressions and expression intensity. Similarly, Oyedotun et al. [21] proposed a deep combined learning method that efficiently learns the features from RGB data and depth map data to recognise facial expressions. Subsequently, Ji et al. [22] introduced a feature fusion network in a cross-category way to identify the facial expressions in cross database scenarios. Hasani et al. [23] proposed a technique for FER in videos using a 3D CNN model. This novel network system was composed of three-dimensional inception-ResNet followed by an LSTM unit that jointly captures spatial relationships between facial structure and temporal information incorporated from various video segments.

Additionally, facial landmark points are utilised as input to the network, emphasising the significance of face elements rather than facial areas that may play a tiny role in producing facial emotions. The proposed approach was tested on four publicly available facial expression databases in subject-independent and cross-database tasks. Asghar et al. [13] proposed a stacked support vector machine ensemble approach for cross-cultural emotion categorisation. The results of SVM were correlated with the probability distributions of the costumes of support vector machines. A naive Bayes predictor makes the ultimate determination regarding the existence of an emotion. The multi-culture dataset is created by combining cross-cultural facial images from JAFFE, TFEID, KDEF, CK+, and Radboud. Fan et al. [24] developed a framework to integrate the discriminative features using convolutional neural networks with handcrafted characteristics such as shape and appearance-based features to enhance the robustness and accuracy of the FER system.

In [25], the authors presented a deep learning method based on the attentional convolutional network capable of focusing on critical areas of the face and outperforms prior models on various datasets, including FER-2013, CK+, FERG and JAFFE. The suggested scheme encodes shape, appearance and extensive dynamic information. Additionally, texture information is retrieved from face parts to improve the texture extraction's discriminative capability. On the CK+ dataset, the framework outperforms state-of-the-art FER techniques.

## 3. Materials and Methods

The proposed facial expression recognition system identifies the seven universal facial expressions (sad, happy, anger, fear, surprise, disgust and neutral). The proposed method first detects the facial region and then extracts the facial area from the input image. The extracted facial region is normalized to achieve consistency and avoid distraction in facial images. The extracted face region is then scaled to suit the network input size after normalization. This pre-processed dataset is used to train the Racial Identity Network (RI-Net) to learn the cultural variations in facial structures. Finally, RIA-DCNN is trained to learn racial features from pre-trained RI-Net and learn expression features directly from the multi-cultural dataset. In the end, the developed model's performance is evaluated using the test set of the multi-cultural dataset. The detailed structural diagram for our proposed method is presented as Figure 2.

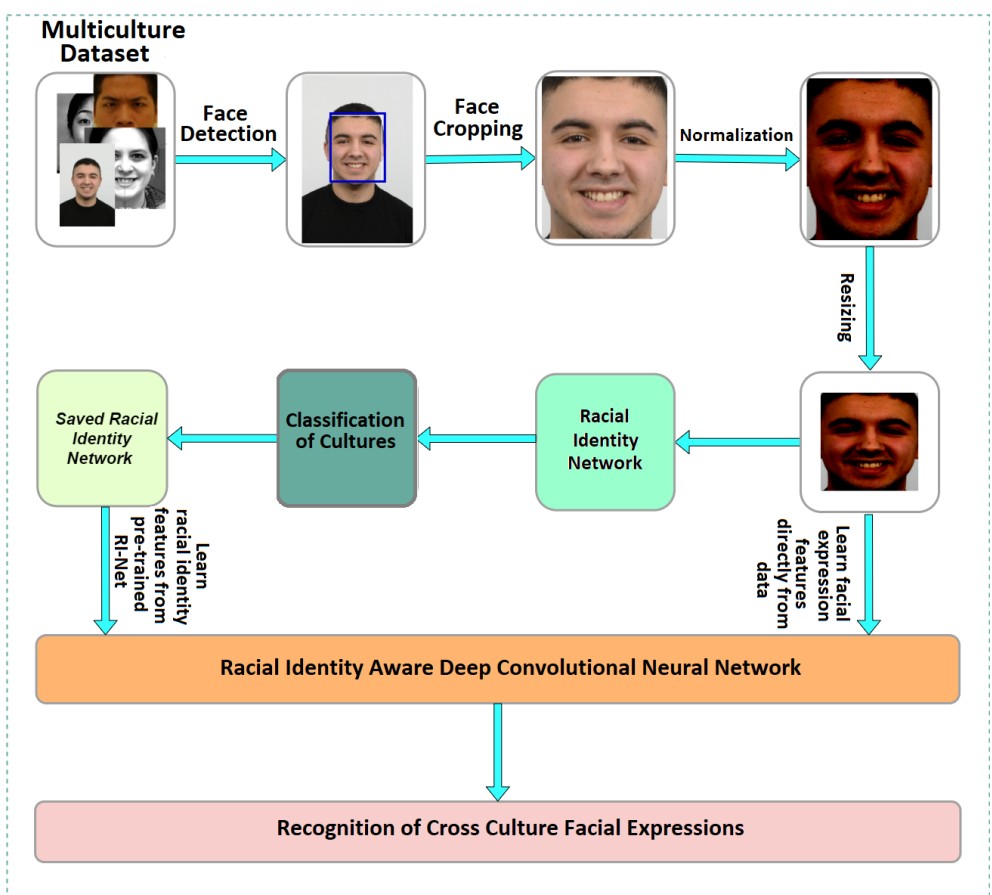

**Figure 2.** Racial identity awareness cross-culture facial expressions recognition framework.

### 3.1. Data Preparation

The multi-cultural facial expression dataset was developed to represent cultural diversity. In his regard, the multi-culture dataset was designed by merging four publicly available facial expression databases (JFFE, TFEID, CK+ and RaFD) derived from five distinct cultures ("Moroccan", "Caucasian", "Taiwanese", "American" and "Japanese") as shown in Figure 1. A detailed description of the multi-culture facial expression database is given in Tables 1 and 2. Each database has its own unique properties, such as the CK+ dataset is the most widely utilized research lab database for FER system evaluation. The CK+ dataset contains 593 posed sequences from 123 individuals, with 304 of them labelled to one of the six distinct emotions: anger, disgust, fear, happiness, sad and surprise. In contrast, JAFFE [26] is a collection of 213 images of ten Japanese women expressing six fundamental emotions: anger, disgust, fear, happiness, sad, surprise and neutral. JAFFE is also a well-known acting database, which indicates that it was created in a controlled

condition. The RaFD is a collection of 67 models (including Caucasian and Moroccan). The RaFD is a high-resolution face database that includes images of eight different emotions. The TFEID database includes 268 images from 40 Taiwanese individuals, each with the six fundamental emotions and neutral expressions.

**Table 1.** Description of multi-culture facial expression database.

| Culture/Race | Gender | Source Dataset | Relevant Expressions |
|---|---|---|---|
| Moroccans | | RaFD | |
| Caucasians | | RaFD | Angry, Disgust, Fear, |
| Taiwanese | Female, Male | TFEID | Happy, Neutral, Sad, |
| Americans | | CK+ | Surprise |
| Japanese | | JAFFE | |

The multi-cultural dataset is presented into two different aspects. First, the dataset is labeled with five cultural representations such as Moroccans, Caucasians, Taiwanese, Americans and Japanese classes to train the RI-Net. Secondly, the multi-cultural dataset is labelled with the seven different facial expressions (happy, sad, surprise, angry, neutral, fear and disgust) to train the RIA-DCNN. The detailed description of the dataset along with facial expressions and number of samples is presented in Tables 1 and 2.

**Table 2.** Numbers of images in the databases used in our studies for seven facial expression classes.

| | CK+ | TFEID | RaFD | JAFFE |
|---|---|---|---|---|
| Happy | 69 | 40 | 75 | 31 |
| Surprise | 82 | 36 | 58 | 30 |
| Sad | 28 | 39 | 67 | 31 |
| Fear | 25 | 40 | 92 | 32 |
| Angry | 45 | 34 | 65 | 30 |
| Disgust | 59 | 40 | 89 | 29 |
| Neutral | 106 | 39 | 73 | 30 |
| **Total** | **414** | **268** | **519** | **213** |

### 3.1.1. Face Detection and Facial Region Extraction

The first step of data preprocessing is to detect the face portion from images. To detect faces from the image, a Viola–Jones face detector is applied. The Viola–Jones face detector uses a set of coordinates to represent the facial area. Thus, face detection assists in determining which portions of an image should be examined to recognize emotions. The Viola–Jones face detector was used to get the set of coordinates of the facial region. By using these coordinates, the rectangular bounded box of the facial region is cropped from the image, as shown in Figure 2. Thus, the extracted facial region of the facial expression image is used for deep learning models training and evaluation. This process eliminates irrelevant information, including background, body and head region.

### 3.1.2. Intensity Normalization and Resizing

After cropping the face region, the facial images normalized to represent uniform intensity. The intensity normalization process minimizes the effect of intensity variations; they significantly undermine the performance of the facial expression recognition system. Figure 2 demonstrates the original image and the normalized image. The size of extracted facial images varies in size. To represent the uniform size of all extracted facial images, the

facial images are resized to 320 × 256 pixels. After performing all preprocess steps, the preprocessed multi-cultural dataset is used to organize a training set and testing set for proposed model training and testing.

### 3.2. Proposed Method for Multi-Cultural Facial Expressions Recognition

Over the last decade, several methods have been developed to address the multiculture FER problems in computer vision. The main focus of these developments is to increase recognition performance. in facial expressions' recognition performance, there are still significant limits in terms of accuracy (poor resolution, size and illumination variation). In this paper, the racial identity-aware deep convolutional neural network architecture is proposed to address these limitations. To develop this model, first, it is required to build a racial identity recognition network. The RI-Net is used as a pre-trained model to extract the racial identity feature from facial expression images. Finally, the RIA-Network jointly learns the racial identity features and facial expression features. This joint learning approach minimizes the effect of cultural variations in facial expression representation in multi-culture facial expression recognition. Thus, let us first see how to build these networks to design the proposed model.

#### 3.2.1. Racial Identity Network

The RI-Net is composed of five convolutional layers. The first three convolution layers are followed by a max-pooling layer, fourth and fifth convolution layer followed by an average pooling layer and flattened layer, respectively. The flattened layer is introduced to transform the feature map matrix to a feature vector which is then fed to the fully connected layer for processing. Finally, a fully connected layer followed by SoftMax to classify the sample facial image as one of the culture identities from Japanese to Taiwanese, Americans, Caucasians, and Moroccans. Here, smaller convolutions are applied to local characteristics while bigger convolutions are applied to global features. The racial identity can be recognized by examining the facial structure, facial appearance and visual representation variations. The nonlinear activation function ReLU is applied to rectify the linear units, which eliminates the vanishing gradient issue associated with certain other activation functions. Figure 3 illustrates the RI-Net configuration. The racial identity network is trained on the multi-cultural dataset. The racial identity network learns facial structure and facial appearance variations in five different cultures (Moroccans, Caucasians, Taiwanese, Americans and Japanese). Because the face anatomy of different cultures differs significantly, it is challenging for a facial expression recognition system to identify facial expressions effectively and reliably in a multi-cultural setting. Later on, the racial identity network is used as a pre-trained network in the proposed RIA-DCNN model.

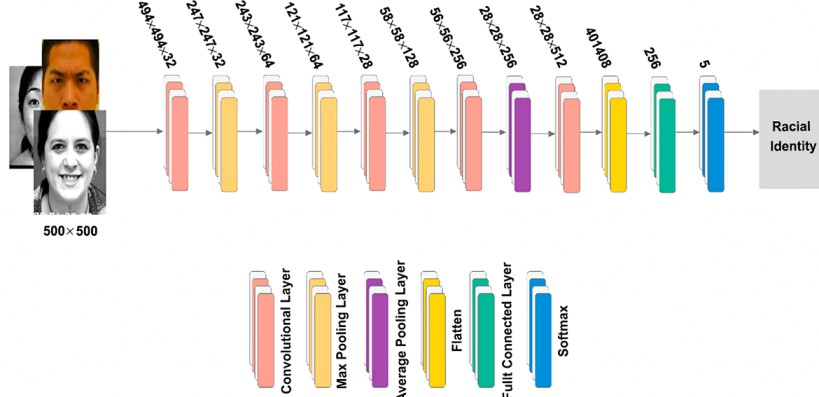

**Figure 3.** The proposed racial identity network (RI-Net) architecture.

### 3.2.2. Racial Identity Aware Deep Convolutional Neural Network

The proposed RIA-DCNN learns facial expression features directly from facial expression images and extracts the racial identity features from a pre-trained racial identity network. The racial identity features work as a supportive feature for cross-culture facial expression recognition. The features of racial identity and facial expression are merged as multi-culture facial expression (MCFE) features and feed these features to subsequent fully connected layers in the RIA-DCNN architecture. By enforcing the marginal independence of facial expression and racial identity, the expression features are expected to be purer and robust to racial identity variations. Figure 4 demonstrates the proposed joint learning approach for racial identity-aware facial expression recognition.

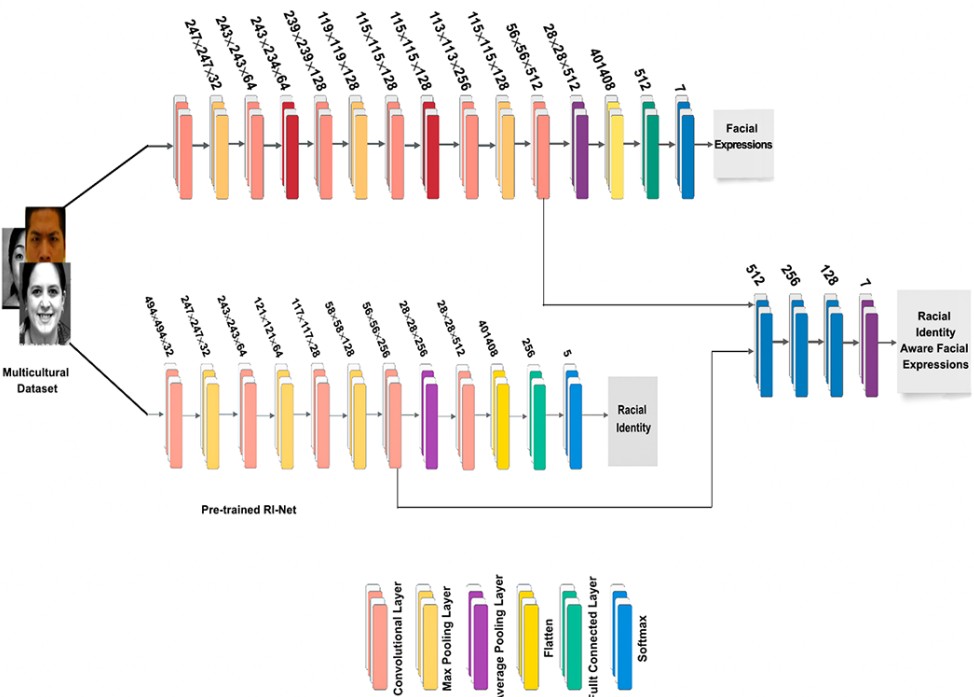

**Figure 4.** The proposed racial identity aware deep convolutional neural network (RIA-DCNN) architecture.

In Figure 4, the upper network is designed using the deep ResNet structure, designed as an external network called ResNet12. It learns facial features from sample facial images. The shortcut connections are applied for deep residual learning and batch normalization layers for fast convergence and high accuracy. The feature maps are not flattened using fully connected layers (FC); the average pooling is applied to produce a more compact representation with fewer parameters. A fully connected layer is applied on concatenated features (racial identity features and facial expressions features). In Figure 4, the top and lower networks are combined into a single joint network. During model training, the facial expression features and racial identity features are concatenated to form a feature vector that is used as input to fully connected layers. The combined network is called the RIA-CNN, which jointly learns facial expressions and racial identity features to recognize multi-cultural facial expressions. The introduction of residual blocks in ResNet architecture has shown remarkable performance in a variety of problems. ResNet is developed on the deep residual learning framework, and optimizing the residual is much simpler than optimizing the original mapping. As a result, ResNet is used to learn the emotion features. With the depth increasing, training to a deep neural network is a difficult task, and sometimes accuracy gets drenched, which leads to more significant training errors. To solve this problem, residual blocks are presented; the difference between residual block and convolution block is the addition of skip connection which carries the input to deeper layers. In Figure 5, input is denoted by x, and f(x) is the desired mapping required by

learning; the dotted box directly learns the mapping f(x) presented on the left side. The architecture of the residual block is presented on the right side; the dotted box portion learns slightly different but easy learning mapping of f(x)-x. The input x is added with the mapping f(x)-x to obtain the actual mapping of f(x). The solid line represents a residual or shortcut connection that adds the value of x to the mapping. This addition of x acts like a residual, hence the name 'residual block'.

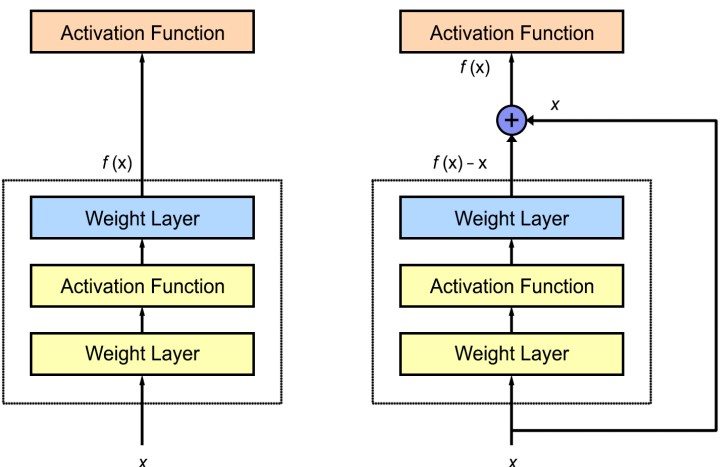

**Figure 5.** Residual network architecture used in racial I.

### 3.3. Performance Evaluation Techniques

To evaluate the performance of the proposed method, the following evaluation metrics have been applied: accuracy, precision, recall and f1 score. Moreover, the confusion matrix used to analyze the performance of models designed in this research. The accuracy can only be useful where each class has the same number of samples. It is worthless if the sample set contains unbalanced class representations. We also applied other evaluation matrices such as precision, recall and f1 score.

## 4. Results and Discussion

The proposed joint deep learning model is trained using a preprocessed multicultural dataset. First, a racial identity network is trained to learn the cultural variations in the multicultural dataset. Later on, the encoding part of the pre-trained RI-Net is connected with the fully connected layer of the RIA-DCNN model. The RIA-DCNN model jointly learns the facial expression features directly from facial images and extracts the racial identity features from pre-trained RI-Net. Prior to model training, the whole dataset has been divided into training and testing sets with the ratio of 70:30. Three different experimental setups were designed. In the first experimental setup, the RIA-DCNN jointly learns the facial expression features and racial identity features. In the second experiment, a simple DCNN model is trained using the facial expression images without racial identity features. In the first two experimental setups, the entire multicultural facial expression dataset is used, whereas, in the third experimental setup, the RAI-DCNN model is trained using the culture specific dataset. The third experimental setup is designed to evaluate the performance of the proposed model on culture specific samples. To evaluate the performance of all experimental setups, different performance evaluation techniques including accuracy, precision, recall, f1-score and confusion matrix were applied.

### 4.1. Results Compiled with Racial Identity Aware (RIA-DCNN) Facial Expressions Recognition

The experimental results indicate that the best accuracy of RIA-DCNN Facial Expression Recognition system achieved is 0.9697. The results for each facial expression by other evaluation matrices are presented in Table 3. From Table 3, it can be seen that the highest percentage of misunderstanding arises between the emotions of angry, sad and fear. Additionally, these findings indicate that, among the seven emotions, happy and

surprise are easier to determine than sad, fear, and angry. These results indicate that best recognition accuracy (100%) is obtained with the emotion happy, while the lowest (92.00%) is achieved with the emotion angry.

**Table 3.** Precision, recall, F1 score and accuracy of RIA-DCNN facial expressions recognition.

| Class Label | Precision | Recall | F1 Score | Accuracy |
|---|---|---|---|---|
| Angry | 0.92 | 0.92 | 0.92 | 0.92 |
| Disgust | 0.96 | 0.96 | 0.96 | 0.96 |
| Fear | 0.92 | 0.98 | 0.95 | 0.98 |
| Happy | 1.00 | 1.00 | 1.00 | 1.00 |
| Neutral | 0.95 | 0.97 | 0.96 | 0.97 |
| Sad | 0.98 | 0.94 | 0.96 | 0.94 |
| Surprise | 1.00 | 0.97 | 0.98 | 0.97 |
| **Average** | **0.96** | **0.96** | **0.96** | **0.96** |

To visualize the performance of the RIA-DCNN model, the confusion matrix for seven facial expressions (angry, disgust, fear, happy, neutral, sad and surprise) is given in Figure 6. The confusion matrix contains a detailed description of each emotion. The *y*-axis indicates the predicted emotion, whereas the *x*-axis denotes the true emotion. The diagonal numbers refer to the correctly classified emotions, whereas the off-diagonal numbers refer to the incorrectly classified emotions. Each row of the matrix contains a clear indication about the strength of confusion between each pair of expressions.

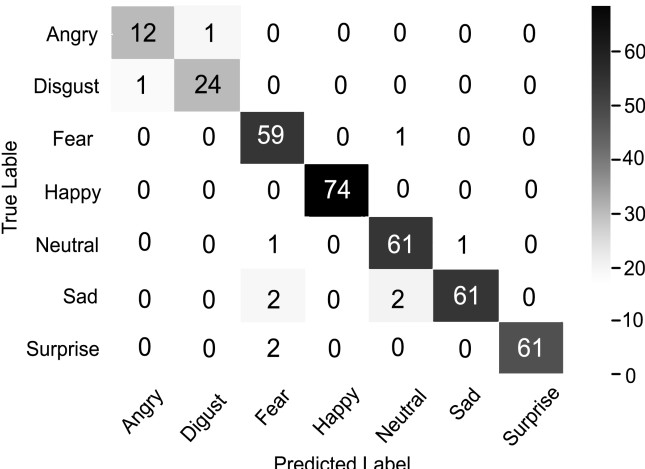

**Figure 6.** Confusion matrix of seven emotions with RIA-DCNN facial expression recognition.

The generated curves for accuracy and loss are shown in Figure 7 after supplementing the whole dataset to the model. RIA-DCNN achieved a minimal error rate after running for 500 hundred epochs. Likewise, Figure 7b illustrates the accuracy score for training and testing using an accuracy curve on the multi-cultural dataset.

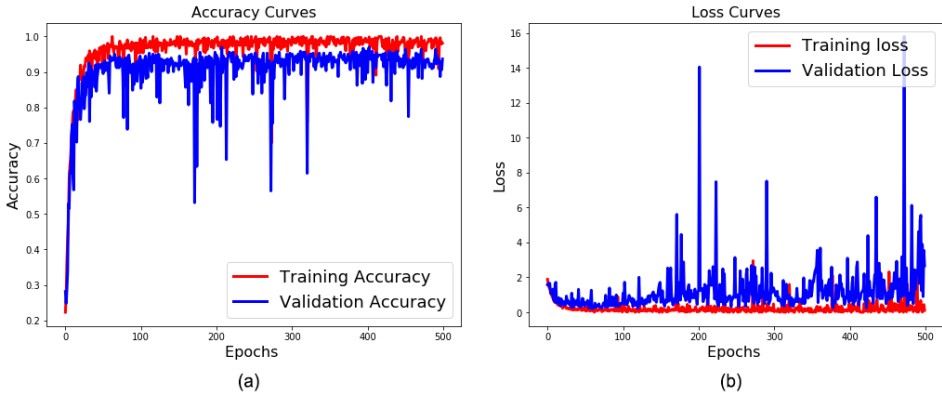

**Figure 7.** (**a**,**b**) Accuracy curve and loss curve with RIA-DCNN facial expression recognition.

*4.2. Results Compiled without Racial Identity Aware Deep Convolutional Neural Network (DCNN)*

The deep convolutional network (DCNN) achieved the average accuracy of 0.93, less than RIA-DCNN's obtained accuracy. Hence, it is likely to be said that the racial identity-aware deep convolutional neural network outperformed the adversary deep convolutional neural network. Table 4 represents the results for all used evaluation metrics, and Figure 8 shows the confusion matrixes for all seven emotions used in this research.

**Table 4.** Precision, recall, F1 score and accuracy of DCNN facial expression recognition.

| Class Label | Precision | Recall | F1 Score | Accuracy |
|---|---|---|---|---|
| Angry | 0.76 | 0.92 | 0.83 | 0.93 |
| Disgust | 0.92 | 0.74 | 0.82 | 0.74 |
| Fear | 0.93 | 0.98 | 0.95 | 0.99 |
| Happy | 1.00 | 0.98 | 0.99 | 0.98 |
| Neutral | 0.97 | 0.94 | 0.96 | 0.95 |
| Sad | 0.97 | 0.95 | 0.96 | 0.96 |
| Surprise | 0.97 | 0.97 | 0.97 | 0.98 |
| **Average** | **0.93** | **0.92** | **0.92** | **0.93** |

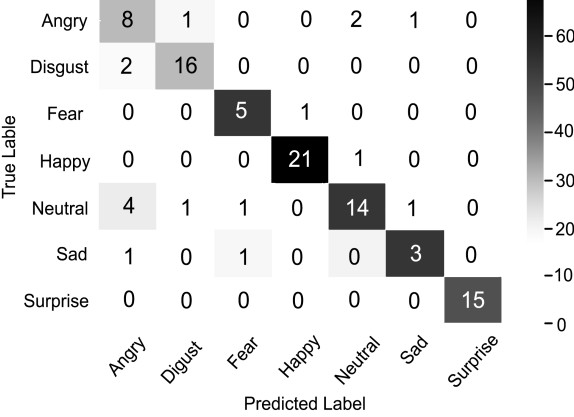

**Figure 8.** Confusion matrix of seven emotions with DCNN facial expression recognition.

Alternatively, on the multiple-cultural dataset, after 500 epochs, DCNN reached a minimal error rate of 0.0672 and a maximum accuracy score obtained of 0.9328, as shown in Figure 9.

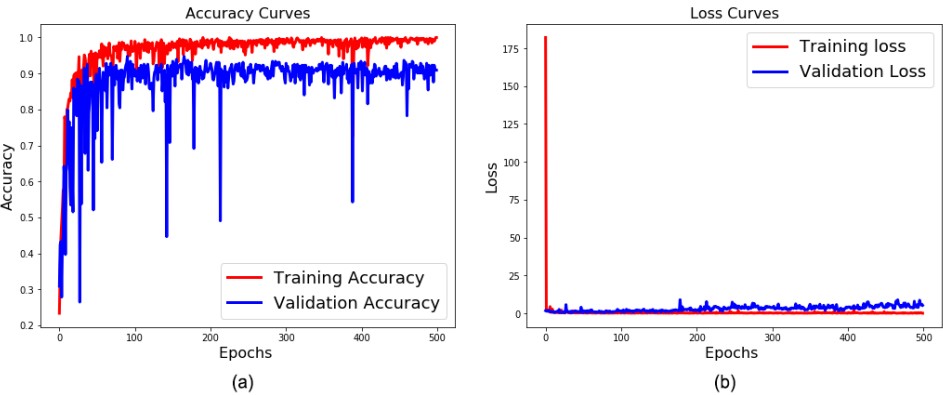

**Figure 9.** (**a**,**b**) Accuracy curve and loss curve with DCNN facial expression recognition.

### 4.3. Culture Wise Facial Expressions Recognition with Racial Identity Aware Deep Convolutional Neural Network (RIA-DCNN)

The culture-wise results of the proposed model for Moroccans, Caucasians, Taiwanese, Americans and Japanese clusters are presented in Table 5 for a detailed description.

The confusion matrixes for five different cultures are presented in Figure 10 from Figure 10a to Figure 10e to demonstrate misunderstandings between seven emotions. Due to significant muscle distortion in face emotion uttering compared to other emotions, happy and surprise emotions are easier to identify. The problematic emotions to detect are fear and sad, while angry emotions are the hardest to detect.

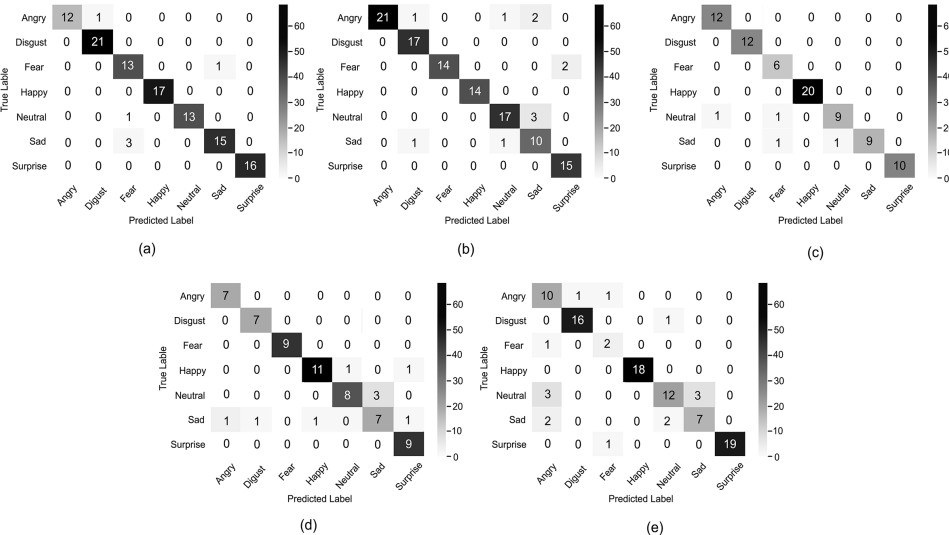

**Figure 10.** Confusion Matrix of multicultural groups namely (**a**) Moroccans; (**b**) Caucasians; (**c**) Taiwanese; (**d**) Americans; and (**e**) Japanese for seven emotions with Deep Convolutional Neural Network Facial Expression Recognition.

To illustrate our proposed model's performance for five distinct cultures (Moroccans, Caucasians, Taiwanese, Americans and Japanese) in Figure 11a–e, the *y*-axis represents the accuracy score, while the *x*-axis represents the number of epochs. The blue curve in Figure 9a represents validation accuracy, whereas the red curve represents training accuracy.

**Table 5.** Precision, recall, F1 score and accuracy score of our proposed method multicultural groups.

| Class Label | Precision | Recall | F1 Score | Accuracy |
|---|---|---|---|---|
| **Moroccans** | | | | |
| Angry | 0.92 | 0.86 | 0.89 | 0.86 |
| Disgust | 0.87 | 1.00 | 0.93 | 1.00 |
| Fear | 0.95 | 0.95 | 0.95 | 0.95 |
| Happy | 1.00 | 1.00 | 1.00 | 1.00 |
| Neutral | 0.95 | 0.95 | 0.95 | 0.95 |
| Sad | 1.00 | 0.94 | 0.97 | 0.94 |
| Surprise | 0.94 | 0.94 | 0.94 | 0.94 |
| **Average** | **0.94** | **0.94** | **0.94** | **0.94** |
| **Caucasians** | | | | |
| Angry | 0.94 | 0.89 | 0.92 | 0.89 |
| Disgust | 0.95 | 1.00 | 0.97 | 1.00 |
| Fear | 1.00 | 0.67 | 0.80 | 0.67 |
| Happy | 1.00 | 0.94 | 0.97 | 0.94 |
| Neutral | 0.82 | 0.78 | 0.80 | 0.78 |
| Sad | 0.65 | 0.87 | 0.74 | 0.87 |
| Surprise | 0.95 | 0.95 | 0.95 | 0.95 |
| **Average** | **0.90** | **0.87** | **0.87** | **0.87** |
| **Taiwanese** | | | | |
| Angry | 1.00 | 1.00 | 1.00 | 1.00 |
| Disgust | 1.00 | 1.00 | 1.00 | 1.00 |
| Fear | 1.00 | 0.94 | 0.97 | 1.00 |
| Happy | 1.00 | 1.00 | 1.00 | 1.00 |
| Neutral | 0.93 | 1.00 | 0.97 | 1.00 |
| Sad | 1.00 | 1.00 | 1.00 | 1.00 |
| Surprise | 1.00 | 1.00 | 1.00 | 1.00 |
| **Average** | **0.99** | **0.99** | **0.99** | **0.99** |
| **Americans** | | | | |
| Angry | 0.53 | 0.66 | 0.59 | 0.67 |
| Disgust | 0.88 | 0.89 | 0.89 | 0.89 |
| Fear | 0.71 | 0.83 | 0.77 | 0.83 |
| Happy | 0.95 | 0.95 | 0.95 | 0.95 |
| Neutral | 0.82 | 0.67 | 0.74 | 0.67 |
| Sad | 0.60 | 0.60 | 0.60 | 0.60 |
| Surprise | 1.00 | 1.00 | 1.00 | 1.00 |
| **Average** | **0.78** | **0.80** | **0.79** | **0.80** |
| **Japanese** | | | | |
| Angry | 0.75 | 1.00 | 0.86 | 1.00 |
| Disgust | 1.00 | 1.00 | 1.00 | 1.00 |
| Fear | 1.00 | 0.83 | 0.91 | 0.83 |
| Happy | 0.75 | 0.60 | 0.67 | 0.60 |
| Neutral | 1.00 | 1.00 | 1.00 | 1.00 |
| Sad | 0.70 | 0.64 | 0.67 | 0.64 |
| Surprise | 0.83 | 1.00 | 0.91 | 1.00 |
| **Average** | **0.86** | **0.86** | **0.86** | **0.86** |

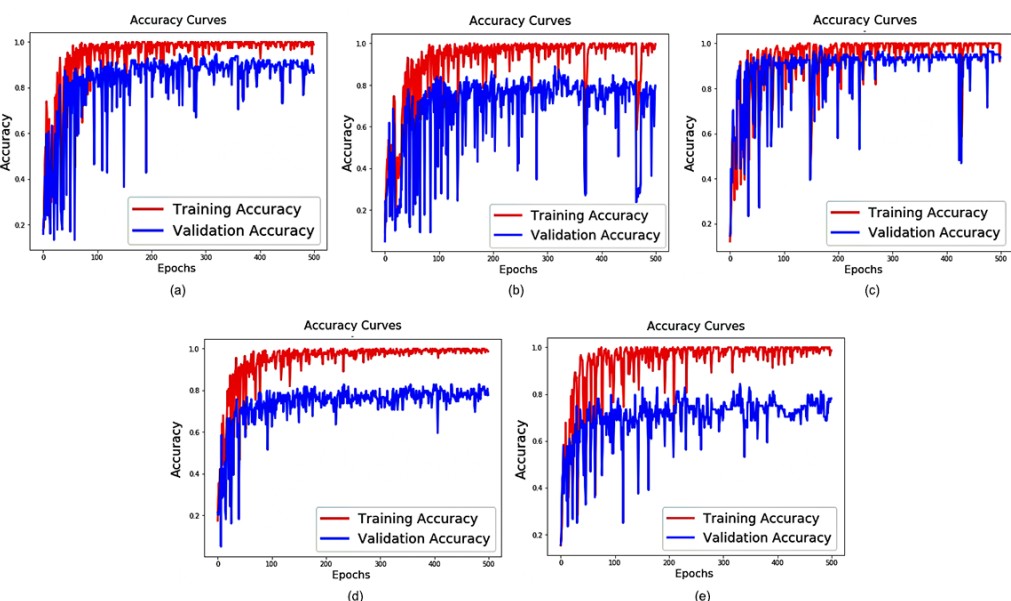

**Figure 11.** Accuracy curves of multicultural groups namely (**a**) Moroccans; (**b**) Caucasians; (**c**) Taiwanese; (**d**) Americans; and (**e**) Japanese for seven emotions with Deep Convolutional Neural Network Facial Expression Recognition.

### 4.4. Discussion and Analysis

The experiments are conducted using two distinct methods: the proposed deep learning method with a racial identity-aware convolutional neural network and a non-racial identity-aware network to show the progression of the suggested method by using racial identity features. More specifically, the proposed method (RIA-DCNN) achieved 96.97% accuracy but with the same conditions without using racial identity features (DCNN) gained 93.28% accuracy. These results express that using racial identity features in cross-culture facial expression recognition improved the results significantly. It also demonstrates that facial expression representation varies across cultures, which affects the performance of the learning algorithm. However, the introduction of racial identity features minimizes the effect of cross cultures' facial structure variations in facial expression representation. The results show that the accuracy of all expressions is not the same. The reason behind the variation in expression recognition accuracy is due to cross-cultural variation in facial structure, facial expression representation and inter-expression resemblance. Langner et al. [27] show the highest average recognition accuracy of emotion happiness and surprise for the RadBoud faces database. These are the most accurately recognizable expressions, as shown in Table 3. It can also be demonstrated that sad and fear emotions are the most difficult to recognize and have the lowest average recognition accuracy, which proves our experimental results' correctness. More specifically, happy is the simplest one to identify and angry is the most confusing one. It occurs because of significant muscle distortion in expressing certain facial expressions as compared to other facial expressions. The culture wise experiments with the racial identity-aware deep convolutional neural network demonstrate that the proposed method achieved outstanding results on Taiwanese culture and performed badly on American culture as compared to other cultures. It demonstrates that the facial structure and illumination variation in Taiwanese objects is low as compared to other cultures. These results also demonstrate that the inter-expression resemblance in Taiwanese objects is very low as compared to Caucasians, Americans and Japanese objects.

### 4.5. Comparison with Existing Approaches

To the best of our knowledge, a limited amount of work has been done on multi-culture facial expression recognition, by combining multiple datasets as a multicultural dataset. Table 6 illustrates the results of other methods that used multiple datasets as a

multicultural dataset. Ali et al. [26] combined the three different facial expression dataset to develop a multicultural dataset for multicultural facial expression recognition. The dataset contains the facial images from Japanese, Taiwanese, Caucasian and Moroccan cultures. However, the proposed RIA-DCNN outperforms the technique presented in [26] with more cultural diversity. Similarly, the work presented in [13] used the multicultural dataset containing facial images from Japanese, Taiwanese, Caucasians, Moroccans, Swedish, Asians, Northern Europeans, Euro-American and Afro-American, achieved 90.70% accuracy. The most recent research in multicultural facial expression recognition is reported in [28] with multicultural facial expression recognition accuracy of 90.1%. The multicultural facial expression recognition technique represented in [13,26,28] used hand crafted features and an ensemble learning approach with a large number of base level classifiers. From Table 6, we can see that the proposed joint deep learning approach outperforms the state-of-the-art multicultural facial expression recognition techniques. The work presented in [13,26,28] used hand crafted feature extraction and feature reduction techniques, whereas the proposed technique works as a single model for feature extraction and classification of multicultural facial expressions. After analyzing these methods, it concludes that the introduction of racial identity features in RIA-DCNN minimizes the effect of cultural variations, facial structure variations and facial expression representation variations in multicultural facial expression recognition. It demonstrates that the racial identity features make the RIA-DCNN more reliable as compared to existing multicultural FER systems.

**Table 6.** Comparison with state-of-the-art models.

| Reference | Author (Year) | Methodology | Dataset | Accuracy |
|---|---|---|---|---|
| [26] | Ali et al. (2016) | boosted NNE (neural network ensemble) | JAFFE, TFEID, and RadBoud | 93.75% |
| [13] | Asghar et al. (2019) | Stacked SVM Ensembles (SVMEs) | Multicultural Dataset | 90.70% |
| [28] | Ali et al. (2020) | Ensemble algorithm | Multicultural Dataset | 90.14% |
| | Proposed Method | RIA-DCNN | Multicultural Dataset | 96.97% |

## 5. Conclusions and Future Work

The current pandemic situation requires online communication with people from diverse cultures around the world. The facial expressions play an important role in expressing the internal emotional state in online conversation. Thus, it is extremely important to develop techniques for efficiently dealing with cross-culture communication hurdles that may arise. In multicultural communication, it is important to clearly understand the internal emotional state with the help of facial expressions to address the variations for clearness and comprehension in order to strengthen the cross-cultural collaboration. We believe that the proposed joint deep learning technique can effectively recognize the multicultural facial expressions while focusing on the effect of cultural variations in facial expression representation. This technique will help to evade the potential communication gap occurring due to cultural diversity. The cultural diversity in facial expression representation requires a new paradigm shift with a higher emphasis on the development of more sophisticated techniques for multicultural facial expression recognition. The RIA-DCNN is capable of learning the multicultural facial expressions with high accuracy, while minimizing the effect of cultural variations, inter-cultural facial expression resemblance and cross-culture facial expression resemblance in facial expression representation. In the future, the proposed model will be extended to recognize the multicultural facial expressions with more cultural diversity.

**Author Contributions:** M.S., G.A. and J.R. have proposed the research conceptualization, methodology and programming. The technical and theoretical framework was prepared by G.A. and J.R. Dataset creation has been performed by I.A. and S.H.A. The technical review and improvement have been performed by M.A.A., A.A.N. and K.M. The overall technical support, guidance, and supervision have been done by G.A. and J.R. The editing and final proofreading have been done by I.A. and J.R. All authors have read and agreed to the published version of the manuscript.

**Funding:** This research work is supported by Data and Artificial Intelligence Scientific Chair at Umm Al-Qura University, Makkah City, Saudi Arabia.

**Institutional Review Board Statement:** Not applicable.

**Informed Consent Statement:** Not applicable.

**Data Availability Statement:** Not applicable.

**Conflicts of Interest:** The authors declare no conflict of interest.

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
