# Peer review of "Racial Identity-Aware Facial Expression Recognition Using Deep Convolutional Neural Networks"

_applsci, doi:10.3390/app12010088_

Round 1

Reviewer 1 Report

The article describes a new racial identity detection method with facial expression recognition by using deep convolutional neural network.

The idea of article is interesting and the results obtained by testing the method with several image well known image databases widely used in scientific literature are similar to those obtained with other similar classifiers.

Minor concerns are related to the following issues: 

The abstract of the article should be re-written in order to describe better the content of the article. Brief description of the methods used in conjunction with image databases and numerically obtained results is necessary.

The text in Fig. 2 should be increased in size. The images in Fig. 3 and Fig. 4 are similar to those in Fig. 2 and should be removed. Fig. 7 that demonstrates the architecture of residual blocks must be completed with more information related to the input/output and other intermediate processing steps. 

Results and Discussion section must be re-written and reduced in size in order to be easy to track and understand. Similar tables and charts should grouped together and those that are redundant must be removed.

Author Response

We are thankful to the reviewers for appreciating the strength of our article. We are also thankful for their valuable comments and suggestions which helped us to improve the manuscript (ID: applsci-1427174) entitled “Racial Identity-aware Facial Expression Recognition using Deep Convolutional Neural Network”. The revised version has been prepared to address the reviewers’ suggestions. We have tried our best to answer their questions here and also included the relevant information in the revised paper as well. The author’s responses to the reviewers’ comments are highlighted here in YELLOW color whereas the actual modifications necessary to be made in the paper are accentuated in YELLOW. Detailed responses to the reviewers’ comments are given below.

Reviewer 1:

The article describes a new racial identity detection method with facial expression recognition by using deep convolutional neural network. The idea of article is interesting and the results obtained by testing the method with several image well known image databases widely used in scientific literature are similar to those obtained with other similar classifiers.

We are thankfull to the respected reviewe for valuable comments.

The abstract of the article should be re-written in order to describe better the content of the article. Brief description of the methods used in conjunction with image databases and numerically obtained results is necessary.

Response: Dear Reviewer, the abstract of the article has been re-written according to your valuable suggestions. [Please see line # 1 to 16 , Abstract]

The text in Fig. 2 should be increased in size. The images in Fig. 3 and Fig. 4 are similar to those in Fig. 2 and should be removed. Fig. 7 that demonstrates the architecture of residual blocks must be completed with more information related to the input/output and other intermediate processing steps. 

Dear Reviewer, The text in Fig. 2 increased in size [Please see figure 2], the Fig. 3 and Fig. 4 removed and Fig. 7 upadted to demonstrate detailed information of residual block. Now Fig. 7 is presented as Fig 5 in the updated Article. The more description has been added as per your direction. [Please see section 3.2.2. Racial Identity Aware Deep Convolutional Neural Network, line # 243 to 254]

Results and Discussion section must be re-written and reduced in size in order to be easy to track and understand. Similar tables and charts should grouped together and those that are redundant must be removed.

Dear Reviewer, we have discussed the results and discussion section in four sections, such as:

4.1. Results Compiled with Racial Identity Aware (RIA-DCNN) Facial Expressions Recognition

4.2. Results Compiled without Racial Identity Aware Deep Convolutional Neural Network (DCNN)

4.3. Culture Wise Facial Expressions Recognition with Racial Identity Aware Deep Convolutional Neural Network (RIA-DCNN)

4.4. Analysis and Discussion

Therefore, the results and discussion section is lengthy.

Thank you very much for your appreciation and valuable suggestions!!!!!

Reviewer 2 Report

The author proposed a racial identify aware deep convolution neural network to recognize the multicultural facial expressions. Four facial expression datasets were combined to prove the effectiveness of joint deep learning approach. Some comments are given as follow:

  • From line 186 to 189 in page 6, the limitation in terms of accuracy and efficiency of facial expressions recognition were demonstrated, and the proposed deep learning network architecture was proposed to address these limitations such as illumination variation, computational cost and geographic complexity and so on. However, I did not find the corresponding method description and result analysis according to those improvements. Please clarify details.
  • In the RID-DCNN architecture in Figure 6, the facial expression network and racial recognition network are jointed. Why did you choose the convolutional layers with 34*26*512 and 37*29*256? Did you try another convolution layer, how did those layers combine?
  • In section 4.3, the facial expressions recognition results for different racial were analyzed and those racial data belong to different dataset. However, in Table 6, the results using proposed method based on mixed dataset were compared with facial expression results using state-of-the-art methods based the different dataset. This comparison does not fully prove the superiority of the novel method.
  • In Figure 6, what is the meaning of the value in the matrixes?
  • All the figures in the manuscript are blurry, please replace the high-quality images.
  • In line 258, there is a spelling error.

Round 2

Reviewer 2 Report

  • From line 189 to 190 in page 6, efficiency of facial expressions recognition about computational cost and geographic complexity haven’t clarified in the modification version. If the computational cost is not your focus, it is recommended to delete this sentence.
  • In Fig.3, some fonts and sizes are different.
